# Effect of Maturation with American Oak Chips on the Volatile and Sensory Profile of a Cabernet Sauvignon Rosé Wine and Its Comparison with Commercial Wines

Miguel Ángel Hernández-Carapia [1], José Ramón Verde-Calvo [1,*], Héctor Bernardo Escalona-Buendía [2] and Araceli Peña-Álvarez [3]

[1] Laboratory of Enology and Fermented Foods, Biotechnology Department, Metropolitan Autonomus University, Mexico City 14387, Mexico; carapiavi@yahoo.com.mx

[2] Sensory and Consumer Laboratory, Biotechnology Department, Metropolitan Autonomous University, Mexico City 14387, Mexico; hbeb@xanum.uam.mx

[3] Department of Analytical Chemistry, Faculty of Chemistry, National Autonomous University of Mexico, Mexico City 04510, Mexico; arpeal@unam.mx

* Correspondence: jrvc@xanum.uam.mx

**Abstract:** Rosé wines are commonly consumed as young wines mainly due to their freshness and fruity character. Nevertheless, in recent years a new market looking for alternatives to traditional wines has emerged. Considering this, the study of the volatile and sensory profiles of a varietal rosé wine aged with oak chips was carried out. Two Cabernet Sauvignon rosé wines were made: one was matured with oak chips and the other without. Both wines were physicochemically characterized. Then, their volatile and sensory profiles were analyzed, also including two commercial wines. The results showed that the produced wines complied with Mexican regulations. Also, they showed greater relative areas in compounds such as ethyl (E)-2-hexenoate, ethyl heptanoate, ethyl nonanoate, ethyl 3-nonenoate, β-citronellol, (±)-*trans*-nerolidol, and β-damascenone. In their sensory profile, they were mostly related to attributes such as berries, prune, bell pepper, and herbaceous notes. Among the compounds related to barrel maturation, only *cis*-oak-lactone was identified in the rosé wine matured with chips. However, it was associated with vanilla, woody, smoky, and spicy attributes. According to the results, the maturation of rosé wines with oak chips could be a good alternative to provide them with unusual notes and thus offer new alternatives to traditional and new wine consumers.

**Keywords:** rosé wine; oak chips; volatile compounds; sensory profile; Cabernet Sauvignon

## 1. Introduction

Wooden barrels began to be used to store and transport wine in European territories under the Roman Empire's dominance more than 2000 years ago. This was to replace the clay containers whose material, in addition to being fragile, was scarcely available in those regions [1]. Since then, and to a greater extent from the 17th century, wood barrels have been used for aging wines [2]. Among the roles of wood in wine aging are the transfer of aromatic and taste compounds to improve the intensity and complexity of wine [3]. However, there are some disadvantages to using wooden barrels for this purpose, for example the high cost of the barrels, their limited lifetime, long aging periods, and the need for a large space in the cellar [4,5]. Diverse techniques have been developed to overcome this situation, either accelerating or replacing the long periods of stay that the aging of wine in wooden barrels implies. One of these techniques is using oak chips during winemaking, which accelerates the transfer of different compounds from wood to wine due to the greater contact surface than the barrel [5].

Depending on the regulations of each country, rosé wine can be defined as a wine that comes from red grapes or a mixture of red and white grapes, fermented in the absence of

solid parts [6], and can be obtained by direct pressing or very short maceration periods (<2 h), by a longer skin maceration (>2 h), using the saignée method, or by blending either the musts or wines from white and red grapes [7]. These kinds of wines are generally consumed as young wines, so their aging in oak barrels or other wood is uncommon. But if they were to be subjected to aging due to their characteristics, their stay in the barrel would be brief at just a few months [8]. An alternative to barrel aging is using oak chips intended to improve the body and introduce wood notes in rosé wines, making the aromatic and flavor profile more complex. The intensity of these notes will depend on aspects such as the level of toasting, the dose used, and the duration of contact with the wine [6]. Although it is true that different research on the use of oak chips as an alternative to barrel aging have been carried out, the vast majority of these have focused on red wine [9–11]. Therefore, the information on the use of this technique on rosé wines is limited.

Talking about the world wine consumption, even though it had ups and downs in the last 20 years, net consumption in 2021 increased by just under 4%, compared to the consumption reported in 2000. Wine consumption in Mexico, including rosé wine, had a much more pronounced increase for the same period, since this value increased from a total consumption of 315,000 hl in 2000 to 1,080,000 hl in 2021 [12], which represents an increase of around 242%. Likewise, the total world consumption of rosé wine grew by around 23% between 2002 and 2019 [13]. In countries like France, the increase in rosé wine consumption is attributed, among other things, to the desire of the Millennial Generation to break the social belief that associates the consumption of red wine with men, and the consumption of rosé wine with women [14]. On the other hand, even though, in Mexico, a recent study shows that the consumption of rosé wine is still mostly by women, the Centennial and Millennial generations have increasingly acquired a taste for this kind of wine in a search for new flavors and new sensations [15]. Therefore, considering the search for new flavors to be an important sector in wine consumption and the advantages offered by using oak chips in wine maturation, this research aimed to contribute to the limited information on aged rosé wines by studying the volatile and sensory profiles of a varietal rosé wine matured with oak wood chips, and by comparing these profiles with those of two commercial rosé wines in order to highlight the differences between the experimental rosé wine and some of the wines available on the Mexican market.

## 2. Materials and Methods

### 2.1. General Description of the Experiment

For this experiment, two Cabernet Sauvignon rosé wines were made. One was maturated with oak chips, and the other without. These wines were physicochemically analyzed. Also, the sensory and volatile profiles of these wines and two commercial Cabernet Sauvignon rosé wines were analyzed to compare them with some existing market products.

#### 2.1.1. Variety and Origin of the Grapes

Grapes of the Cabernet Sauvignon variety were used to elaborate the wine. They were obtained during the harvest of September 2015 in the vineyards of the "Cuna de Tierra" winery, located in Dolores Hidalgo, Guanajuato, Mexico (21°12′29″ N 100°51′ 08″ W). This municipality has a semiarid climate with maximum and minimum temperatures of 36.5 °C and 3.8 °C, respectively. The average annual precipitation is 564.1 mm [16]. In the vintage year, the maximum and minimum temperatures were 30.3 °C and 6.3 °C, respectively. The average annual precipitation was 632.2 mm [17]. The training system used for growing the vines was espalier, and the vine pruning was double cordon.

This variety was used because it is one of the most cultivated in the world [18], in such a way that the wine could be replicated in many regions.

#### 2.1.2. Winemaking Process

The must used to make the rosé wine was obtained through 3 h skin maceration at 10 °C. Then, the solid grape parts were pressed. Sulfites were added to the must at a

concentration of 100 ppm, and bentonite was added at 200 ppm, which was performed at 5 °C for 24 h. Afterward, the must was inoculated and fermented at 20 °C for 12 days in a 20 L tank using a *Saccharomyces bayanus* yeast strain (Lalvin EC-1118, Lallemand), due to its lower production of acetic acid and ethyl acetate compared to *S. cerevisiae* [19]. The yeast was previously rehydrated for 20 min in 10 times its weight of purified water and was subsequently added to the must. Once the fermentation was finished, the wine was racked to eliminate the lees, and then free sulfur dioxide was adjusted to a concentration of 50 ppm. The wine with sulphites was stored at 5 °C for 30 days for clarification and tartaric stabilization. Finally, dregs were removed by racking the wine again, and immediately the wine was bottled in 750 mL green glass bottles, half without oak chips (the control wine), and the other half (the experimental wine) in bottles containing heavy toasted American white oak chips (Barricas Cobar, Tlalmanalco, Edo. Mex., Mexico) at a concentration of 3 g per liter of wine. The chip dimensions were 1.2 × 1.0 × 0.8 cm. The contact time of the wine with the chips was 28 days and the maturation temperature was 20 °C. Next, the samples were analyzed. Both the number of chips and the contact time were chosen based on a study carried out on red wine by Espitia-López [20].

### 2.2. Physicochemical Characterization of Wines

All physicochemycal analyses were made in triplicate.

### 2.2.1. pH

The method 9.2.7, as reported by Jacobson [21], was used for measuring pH.

An amount of 50 mL of wine was added to a 100 mL graduated beaker containing a stir bar. This was then placed on a stir plate, stirring at slow speed was activated, and the pH was measured with a potentiometer (Conductronic PC45).

### 2.2.2. Titratable Acidity

The method 9.2.8, as reported by Jacobson [21], was used for measuring titratable acidity.

An amount of 25 mL of wine and 25 mL of distilled water were added to a 250 mL graduated beaker containing a stir bar. This beaker was placed on a stir plate, stirring at low speed was activated, and titration was carried out with a 0.1 N NaOH solution until reaching a pH of 8.2, which was measured with a previously calibrated potentiometer (Conductronic PC45).

### 2.2.3. Reducing Sugars

For the measurement of reducing sugars, the Lane-Eynon method, as described by the ASBC [22], with some modifications, was used.

A 25 mL aliquot of wine was taken and a 1:4 (*v/v*) dilution in water was carried out. Subsequently, to this diluted solution lead subacetate was added to decolorize, and then it was subjected to filtration. Once filtered, this solution was used for the titration of 5 mL of Fehling's reagent diluted with 10 mL of distilled water and previously standardized with a 0.5% (*w/v*) dextrose solution.

### 2.2.4. Free and Total Sulfur Dioxide

The Ripper method, described in 9.2.12 by Jacobson [21], was used for measuring free and sulfur dioxide.

Free $SO_2$: amounts of 25 mL of wine, 5 mL of $H_2SO_4$ (dilution 1:3 with distilled water), and 5 mL of 1% *w/v* starch solution were added to a 250 mL Erlenmeyer flask. The mixture was homogenized by gentle shaking. This was immediately titrated with 0.02 N iodine solution.

Total $SO_2$: In a 250 mL Erlenmeyer flask, 25 mL of wine and 25 mL of 1 N NaOH solution were mixed. The flask was stoppered and allowed to incubate at room temperature for 10 min. Afterwards, the analysis proceeded in the same way as for the free $SO_2$.

### 2.2.5. Volatile Acidity

The method 9.2.17, as reported by Jacobson [21], with some modifications, was used for measuring the volatile acidity of wine, but a micro Kjeldahl apparatus was used instead of cash still.

An amount of 5 mL of wine was placed in a micro Kjeldahl apparatus and distillation was maintained until 20 mL of distillate was captured in a 125 mL Erlenmeyer flask containing 10 mL of cold distilled water. A few drops of phenolphthalein were added as an indicator, and it was titrated with a 0.01 N NaOH solution.

### 2.2.6. Total Anthocyanins Content

The pH differential method, as described by Giusti and Wrolstad [23], was used to measure anthocyanin content.

Two 5 mL aliquots of wine were taken and each one was added to a 25 mL volumetric flask. One of the flasks was brought to volume with pH = 1.0 buffer and the other one with pH = 4.5 buffer. The flasks were left for 15 min to reach equilibrium, and then the absorbance was read in a spectrophotometer (Thermo Spectronic, BioMate 3 Model) at a wavelength of 510 nm, as well as at 700 nm to correct the effect of sample turbidity. Distilled water was used as a blank to perform the readings. The anthocyanins content was calculated by using the formula reported by Giusti and Wrolstad [23].

### 2.2.7. Alcohol Content

The method described by the ASBC [24] was used for measuring alcohol content.

In a distillation flask, 50 mL of wine and 25 mL of distilled water were added. This solution was subjected to distillation from which approximately 48 mL of the distillate was collected directly in a 50 mL volumetric flask. The flask was brought to the mark with distilled water, the specific gravity was determined by pycnometry, and, consulting reference tables, the ethanol content in the wine was estimated.

### 2.2.8. Total Phenolics Content Determination

The total phenolics content was analyzed using the method reported by Singleton and Rossi [25], but ten times fewer reagents and samples were used.

In a 10 mL volumetric flask containing 7.5 mL of distilled water, 100 μL of wine and 300 μL of Folin's reagent were mixed. Next, 1 mL of a 20% $w/v$ anhydrous $Na_2CO_3$ solution was added, and the volume was brought to volume with distilled water and mixed thoroughly. This solution was incubated for one hour, mixed thoroughly, and then absorbance at 675 nm was measured. Distilled water was used instead of wine in a blank solution. The total polyphenol content was calculated using a calibration curve of gallic acid in the range of 120 to 1200 mg/L.

### 2.3. Volatile Compounds Analysis

Analyses of volatile compounds were made in triplicate for each sample.

### 2.3.1. Headspace Solid-Phase Micro Extraction Procedure (HS-SPME)

The volatile compounds extraction of the wines was carried out as follows: an amount of 2.5 mL of wine was pipetted into an 11 mL HS-transparent-glass vial (Sigma–Aldrich, Saint Louis, MI, USA) containing 2.5 mL of distilled water, 0.8 g of NaCl, and a magnetic stirrer bar. Immediately, the vial was capped and placed in a 40 °C water bath. Stirring was applied at 1200 rpm, and an equilibrium time of 10 min was left. Next, a DVB/PDMS/CAR fiber (Supleco, Bellafonte, PA, USA), previously conditioned as recommended by the manufacturer, was exposed in the headspace vial for 60 min to extract the analytes. Finally, the fiber was desorbed for 10 min into the GC injection port. The fiber was reconditioned at 250 °C for 10 min after each injection.

2.3.2. Volatile Compounds Identification

The volatile analysis was carried out using an HP-5890 GC equipped with a mass spectrometer detector (MS HP-5971) and a Zebron ZB5 column (30 m length × 0.25 mm i.d., 0.25 μm film thickness). The injector was operated in splitless mode at 250 °C and the transfer line at 280 °C. To separate the volatile compounds, an oven temperature program was set. The initial temperature of the oven was 40 °C for 1 min, then the temperature was increased to 185 °C at a rate of 5 °C/min. Helium was used as carrier gas at a constant 1 mL/min flow. The volatile compounds were identified using a mass spectral database (G1701DA MSD Chem Station Build D. 00.00.38 Standard MSD Version) and by standards injection.

### 2.4. Sensory Analysis

The rosé wine aged with oak chips (RWCH wine), the rosé wine without oak chips (RW wine), and two commonly available commercial wines in Mexican liquor stores (also from the 2015 vintage and produced by pre-fermentative maceration)—one from Baja California, Mexico (wine C1) and one from Valle Central, Chile (wine C2)—were evaluated through a quantitative descriptive profile. For this purpose, seven judges were trained as reported by Hernández-Carapia et al. [26]. They evaluated the different visual, olfactory, and flavor attributes of the wines by placing a mark on a 15 cm continuous line [27]. This test was performed using Fizz software (Version 2.5; Biosystems, Courtenon, France).

### 2.5. Statistical Analysis

The data obtained from the physicochemical analysis of the wines RW and RWCH, and the volatile analysis of the wines RW, RWCH, and the two commercial wines, were analyzed through the ANOVA test to evaluate the differences between them. This test was carried out with the statistical package NCSS 2020 Version 20.0.3.

The data obtained from the sensory analysis and those obtained from the volatile analysis were subjected to a Principal Component Analysis to obtain the sensory descriptors and the volatile compounds that were mostly associated with each of the wines. The XLSTAT statistical package (Version 2018.7, XLSTAT-Sensory package, Addinsoft, Paris, France) was used for this analysis.

### 3. Results and Discussion

#### 3.1. Physicochemical Properties Analysis

Table 1 shows the physicochemical properties of the experimental and the control rosé wines. According to these, both the rosé wine with chips and the rosé wine without chips comply with the parameters established for wines by the Mexican regulation NOM-199-SCFI-2017 [28] as well as with the limits established by the OIV for $SO_2$ and volatile acidity [29]. In agreement with the Mexican regulation, the obtained experimental wines are classified as medium dry wines, since it considers a range between 4.1 and 12.0 g of reducing sugars per liter for this classification. However, considering the parameters established by the International Organization of Vineyard and Wine (OIV), the rosé experimental wines are classified as dry, since the OIV sets a reducing sugar content of up to 9 g/L when this value is not greater than 2 g/L concerning the total acidity content, expressed in grams of tartaric acid per liter [30].

The analysis of variance results (ANOVA) showed that, among the measured parameters in the experimental and the control wines, there was no significant difference in the content of alcohol, reducing sugars, volatile acidity, and pH. However, the RW wine had higher titratable acidity, total anthocyanin, and total and free $SO_2$ values. On the other hand, the RWCH wine had a higher total phenolics content.

**Table 1.** Physicochemical properties of experimental and commercial rosé wines.

| Property | Rosé Wine without Chips | Rosé Wine Aged with Chips | C1 | C2 |
|---|---|---|---|---|
| pH | $3.37 \pm 0.03$ [a] | $3.36 \pm 0.03$ [a] | $3.22 \pm 0.01$ [b] | $2.98 \pm 0.01$ [c] |
| Alcohol (% *v/v*) | $12.47 \pm 0.23$ [a] | $12.76 \pm 0.29$ [a] | $12.41 \pm 0.47$ [a] | $12.38 \pm 0.29$ [a] |
| Reducing sugars (g/L) | $6.33 \pm 0.17$ [a] | $6.34 \pm 0.30$ [a] | $4.63 \pm 0.11$ [b] | $4.46 \pm 0.06$ [b] |
| Titratable acidity (g of tartaric acid/L) | $5.77 \pm 0.15$ [b] | $5.47 \pm 0.05$ [c] | $6.4 \pm 0.03$ [a] | $5.74 \pm 0.03$ [b] |
| Volatile acidity (g of acetic acid/L) | $0.76 \pm 0.03$ [a] | $0.70 \pm 0.04$ [a] | $0.28 \pm 0.01$ [c] | $0.34 \pm 0.01$ [b] |
| Total $SO_2$ (mg/L) | $61.46 \pm 1.76$ [b] | $52.76 \pm 0.87$ [c] | $95.57 \pm 1.47$ [a] | $104.96 \pm 6.77$ [a] |
| Free $SO_2$ (mg/L) | $29.79 \pm 0.88$ [b] | $23.58 \pm 1.76$ [d] | $26.02 \pm 1.95$ [c] | $34.08 \pm 1.52$ [a] |
| Total anthocyanin (mg of malvidin-3-glucoside/L) | $62.78 \pm 2.11$ [a] | $58.16 \pm 1.49$ [b] | $4.55 \pm 0.13$ [d] | $6.34 \pm 0.08$ [c] |
| Total phenolics (mg/L) | $349.4 \pm 5.0$ [b] | $394.2 \pm 9.0$ [a] | $168.5 \pm 0.98$ [c] | $161.44 \pm 2.67$ [d] |

Analyses were made in triplicate. Different letters indicate significant differences between wines according to ANOVA test.

Regarding the commercial wines, both had a lower reducing sugar content than the experimental wines. Despite this, they are also classified as medium dry wines by Mexican regulations, and as dry according to the OIV classification. Also, wines C1 and C2 had a considerably lower concentration of total anthocyanins and total phenols than the experimental wines, probably due to differences in maceration times or the content of these compounds in the berries used for each wine. On the other hand, the commercial wines had a higher total sulfur content than the experimental wines.

Concerning the lower levels of the wine aged with chips in the titratable acidity, Călugăr et al. [31] reported a decrease in this parameter in a white wine aged with chips. This effect could be due to some compounds extracted from the oak chips that may combine with organic acids and form volatile compounds, which causes a decrease in titratable acidity [32]. Regarding anthocyanin content, Jordão et al. [33] reported a decrease in color intensity as well as in anthocyanin content in synthetic wine solutions. This decrease could be attributed to reactions between anthocyanins and ellagitannins extracted from the wood [34]. In addition to the anthocyanin decrease, the tannins extracted from the oak chips could also be responsible for the decrease in both free and total $SO_2$ since, in a study carried out by Ugliano et al. [35], both condensed and hydrolyzable tannins were associated with $SO_2$ consumption. At the same time, these tannins extracted from the oak chips seem to be also responsible for the higher values of the RWCH in the total phenolics content. Regarding this, Psarra et al. [36] reported an increase in the total phenolics of a model wine enriched with oak chips. Similarly, Santos et al. [37] reported higher values of total phenolics in rosé wines aged with chips from different wood species, among them being American oak chips.

*3.2. Volatile Analysis*

Forty-one compounds were identified: twenty-four esters, five alcohols, three terpenes, three aldehydes, three norisoprenoids, one ketone, one lactone, and one acetal. However, not all of them were present in the four analyzed wines. As shown in Table 2, methyl esters of octanoic and decanoic acid could be identified only in wine C1. In the same way, three compounds were only detected in one wine: benzaldehyde in wine C2, 1-(1-ethoxyethoxy)-pentane in RW wine, and *cis*-oak-lactone in RWCH wine. Ethyl 9-decenoate, diethyl glutarate, vinyl decanoate, β-linallol, and 1,1,6-trimethyl-1,2-dihydronaphthalene (TDN) were only identified in the two commercial wines. Similarly, ethyl-3-nonenoate, ethyl 3-hydroxytridecanoate, decanal, and β-damascenone were only identified in the two experimental rosé wines.

**Table 2.** Relative areas of volatile compounds identified in experimental and commercial wines.

| Volatile Compound | RWCH | RW | C1 | C2 |
|---|---|---|---|---|
| **Esters** | | | | |
| Ethyl 2-furancarboxylate | 38.71 [b] | 27.02 [b] | 41.05 [b] | 100 [a] |
| Propanoic acid, 2-methyl-, 1-(1,1-dimethylethyl)-2-methyl-1,3-propanediyl ester | 100 [a] | 92.49 [a] | 94.50 [a] | 63.08 [a] |
| Ethyl hexanoate | 64.18 [a] | 65.08 [a] | 100 [a] | 77.16 [a] |
| Hexyl acetate | 32.63 [b] | 31.79 [b] | 68.36 [a] | 100 [a] |
| Ethyl (E)-2-hexenoate | 100 [a] | 99.45 [a] | 49.46 [c] | 72.20 [b] |
| Ethyl heptanoate | 97.47 [a] | 100 [a] | 42.27 [b] | N.D. |
| Diethyl succinate | 16.21 [c] | 15.01 [c] | 31.80 [b] | 100 [a] |
| Methyl octanoate | N.D. | N.D. | 100 [a] | N.D. |
| Ethyl octanoate | 38.49 [c] | 38.31 [c] | 100 [a] | 63.01 [b] |
| Isopentyl hexanoate | 74.72 [b] | 76.77 [b] | 100 [a] | 79.81 [b] |
| 2-phenylethyl acetate | 42.37 [b] | 34.08 [b] | 39.73 [b] | 100 [a] |
| Ethyl nonanoate | 100 [a] | 87.90 [a] | 65.02 [b] | N.D. |
| Ethyl 3-nonenoate | 76.40 [a] | 100 [a] | N.D. | N.D. |
| Ethyl decanoate | 28.65 [c] | 29.22 [c] | 100 [a] | 53.50 [b] |
| Methyl decanoate | N.D. | N.D. | 100 [a] | N.D. |
| 2-methylpropyl octanoate | 100 [a] | 85.68 [a] | 75.11 [a] | 40.57 [b] |
| 3-methylbutyl octanoate | 82.57 [a] | 78.70 [a] | 100 [a] | 86.81 [a] |
| Ethyl hydroxycinnamate | 100 [a] | 84.34 [a] | 29.00 [b] | N.D. |
| Ethyl dodecanoate | 69.44 [b] | 72.42 [b] | 100 [a] | 50.09 [b] |
| Ethyl phenylacetate | 42.13 [b] | 40.60 [b] | 39.06 [b] | 100 [a] |
| Ethyl 9-decenoate | N.D. | N.D. | 100 [a] | 66.35 [b] |
| Ethyl 3-hydroxytridecanoate | 89.52 [a] | 100 [a] | N.D. | N.D. |
| Diethyl glutarate | N.D. | N.D. | 69.70 [b] | 100 [a] |
| Vinyl decanoate | N.D. | N.D. | 57.41 [b] | 100 [a] |
| **Alcohols** | | | | |
| 2-ethyl-1-hexanol | 75.21 [a] | 84.86 [a] | N.D. | 100 [a] |
| 2-Phenylethanol | 76.08 [b] | 63.49 [b] | 25.63 [c] | 100 [a] |
| 1-nonanol | 100 [a] | 85.14 [b] | 15.27 [c] | N.D. |
| 1-dodecanol | 65.46 [b] | 65.38 [b] | 57.43 [b] | 100 [a] |
| Phenol, 2,4-bis(1,1-dimethylethyl)- | 11.86 [b] | 8.22 [b] | N.D. | 100 [a] |
| **Terpenes** | | | | |
| β-citronellol | 92.84 [a] | 100 [a] | 35.87 [b] | N.D. |
| (±)-*trans*-Nerolidol | 100 [a] | 88.74 [a] | 31.14 [b] | N.D. |
| β-linalool | N.D. | N.D. | 81.80 [a] | 100 [a] |
| **Aldehydes** | | | | |
| Nonanal | 100 [a] | 78.85 [a] | 19.96 [b] | 25.51 [b] |
| Decanal | 98.08 [a] | 100 [a] | N.D. | N.D. |
| Benzaldehyde | N.D. | N.D. | N.D. | 100 [a] |
| **Lactones** | | | | |
| *Cis*-oak-lactone | 100 [a] | N.D. | N.D. | N.D. |
| **Norisoprenoids** | | | | |
| β-damascenone | 100 [a] | 87.17 [a] | N.D. | N.D. |
| *trans*-Geranylacetone | 100 [a] | 86.69 [a] | 42.28 [b] | 92.38 [a] |
| Naphthalene, 1,2-dihydro-1,1,6-trimethyl- | N.D. | N.D. | 100 [a] | 83.77 [a] |
| **Ketones** | | | | |
| 2-nonanone | N.D. | N.D. | 100 | N.D. |
| **Acetals** | | | | |
| 1-(1-ethoxyethoxy)-pentane | N.D. | 100 | N.D. | N.D. |

N.D.= not detected; RWCH = Rosé wine with oak chips; RW = Rosé wine without chips; C1 = Commercial wine 1; C2 = Commercial wine 2. Analyses were made in triplicate. Different letters indicate significant differences between wines according to ANOVA test.

The same volatile compounds were detected in the two experimental rosé wines, except for 1-(1-ethoxyethoxy)-pentane, which was only detected in the RW wine, and *cis*-oak lactone, which was only detected in RWCH wine because it is extracted from wood

oak. There was no significant difference in the relative areas of these compounds, except for 1-nonanol, which was significantly higher in RWCH wine than in RW wine.

Among the four wines analyzed, RW and RWCH wines had significantly higher relative areas in twelve volatile compounds (six esters, one alcohol, two terpenes, two aldehydes, and one norisoprenoid). The C2 was the commercial wine with significantly higher relative areas in thirteen compounds (seven esters, three alcohols, one terpene, one aldehyde, and one norisoprenoid). Generally, terpenes and norisoprenoids are classified as primary or varietal aroma compounds [38]. According to Lee et al. [39], the concentration of norisoprenoids in grapes could be related to factors such as sun exposure, defoliation of the vine, and the microclimate of the bunch. Regarding the content in wine, the concentration of these compounds could be associated, in addition to the concentration in the grapes, with oenological practices, such as maceration conditions [40]. Therefore, the presence of β-damascenone only in the experimental rosé wines and of TDN only in the commercial wines could be related to the differences in agricultural practices, the climatic conditions in which the grapes were produced, and the differences in winemaking processes.

On the other hand, most of the compounds identified in the wines studied are considered fermentation metabolites or secondary aromas, according to Romano et al. [38]. Among these compounds are esters, alcohols, and carbonyl compounds (aldehydes and ketones). Although grapes contain many of these volatile compounds, mainly esters, their concentration is considered negligible compared to wine. Most odorous wine esters are formed during fermentation and storage via the enzymatic or non-enzymatic esterification of carboxylic acids and the correspondent alcohol [41]. Since ethanol is the most abundant alcohol in wine, most esters formed in wine are ethyl esters [41], in which the alcohol group is ethanol and the acid group is a medium-chain fatty acid formed in yeast lipid metabolism [42]. As shown in Table 2, most of the esters identified in the present investigation belong to this group. Similarly, acetyl-CoA is the most abundant acyl-CoA in an alcoholic *Saccharomyces cerevisiae* fermentation [42] and, although it is not specified for *S. bayanus*, it could be similar because yeast acetyl-CoA is generated mainly in the mitochondria by the decarboxylation of pyruvate [43]. Thus, acetate esters (in which the acid group is acetate, and the alcohol group is either ethanol or complex alcohol derived from amino acid metabolism) constitute the other main ester categories [32]. The differences in the relative areas of these compounds may be related to the diverse composition of the musts used to obtain the wines and the conditions used in the fermentation stage. The composition of the must, like its nitrogen compounds content [44], the fermentation temperature [45], and the yeast strain used [46], are among the reported factors that can affect the volatile profile of the wine.

Finally, 1-(1-ethoxyethoxy)-pentane (the only acetal identified) as well as (*cis*)-oak-lactone (the only oak-derived compound identified) are considered tertiary aromas related to wine aging [38]. Acetals are products of condensation reactions between aldehydes and alcohols [47]. As mentioned, the compound (*cis*)-oak-lactone was only identified in RWCH wine because it was the only one in contact with oak wood. This lactone and (*trans*)-oak-lactone are the most significant lactones extractable from oak [48] and their increase is favored by roasting [49]. However, the *cis* configuration is the more prevalent form in wine and has a lower sensory threshold than the *trans* isomer (0.074 mg/L for the *cis* isomer versus 0.32 mg/L for the *trans* isomer), thus it is considered a stronger odorant [50].

Figure 1 shows the chromatogram of the RWCH wine with the 31 volatile compounds that were identified. Ethyl hexanoate, 2-phenylethanol, ethyl octanoate, and ethyl decanoate were the compounds whose peaks showed a greater abundance. On the contrary, ethyl hydroxycinnamate, 2-ethyl-1-hexanol, ethyl 3-hydroxytridecanoate, and (*cis*)-oak-lactone showed a lower signal intensity.

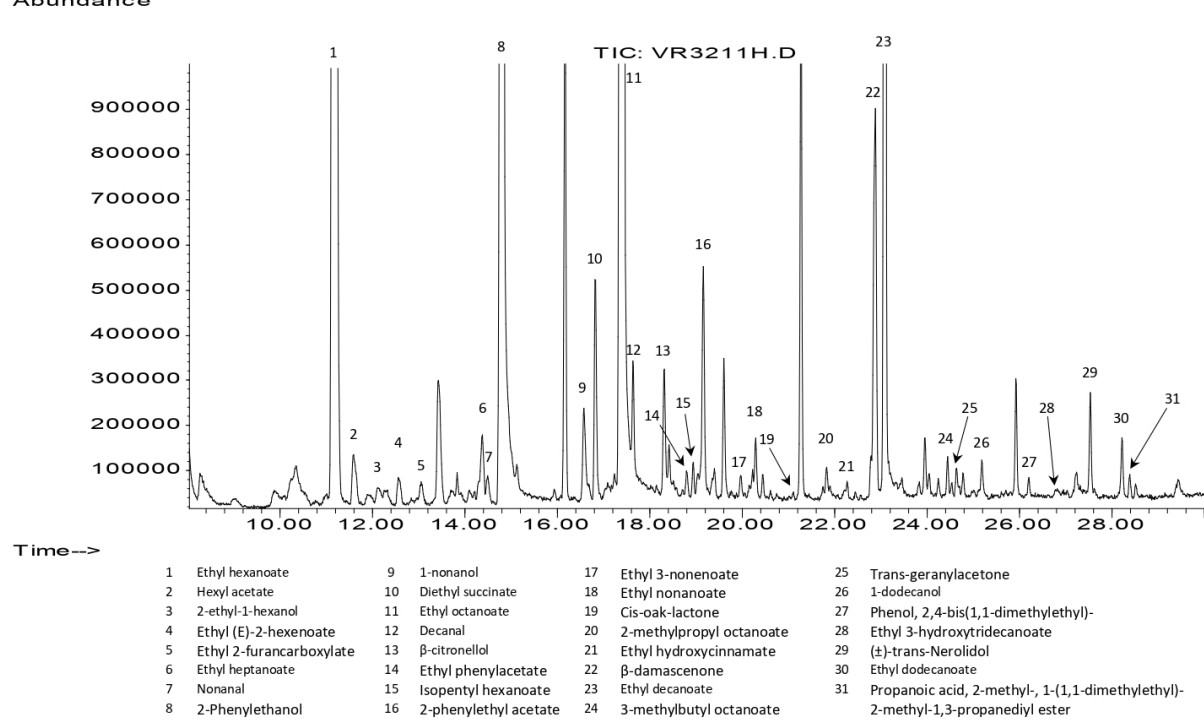

**Figure 1.** HS-SPME-GC-MS chromatogram of the rosé experimental wine matured for 28 days with 3 g of oak chips (RWCH).

| | | | | | | | |
|---|---|---|---|---|---|---|---|
| 1 | Ethyl hexanoate | 9 | 1-nonanol | 17 | Ethyl 3-nonenoate | 25 | Trans-geranylacetone |
| 2 | Hexyl acetate | 10 | Diethyl succinate | 18 | Ethyl nonanoate | 26 | 1-dodecanol |
| 3 | 2-ethyl-1-hexanol | 11 | Ethyl octanoate | 19 | Cis-oak-lactone | 27 | Phenol, 2,4-bis(1,1-dimethylethyl)- |
| 4 | Ethyl (E)-2-hexenoate | 12 | Decanal | 20 | 2-methylpropyl octanoate | 28 | Ethyl 3-hydroxytridecanoate |
| 5 | Ethyl 2-furancarboxylate | 13 | β-citronellol | 21 | Ethyl hydroxycinnamate | 29 | (±)-trans-Nerolidol |
| 6 | Ethyl heptanoate | 14 | Ethyl phenylacetate | 22 | β-damascenone | 30 | Ethyl dodecanoate |
| 7 | Nonanal | 15 | Isopentyl hexanoate | 23 | Ethyl decanoate | 31 | Propanoic acid, 2-methyl-, 1-(1,1-dimethylethyl)- |
| 8 | 2-Phenylethanol | 16 | 2-phenylethyl acetate | 24 | 3-methylbutyl octanoate | | 2-methyl-1,3-propanediyl ester |

### 3.3. Sensory Analysis Results

The results of the principal components analysis showed that RW wine was associated with odor attributes such as apple, prune, yeast, herbaceous, bell pepper, and spicy. The RW wine was also related to flavor descriptors such as yeast, spicy, herbaceous, and butter. The latter could be linked to the presence of propanoic acid, 2-methyl-, 1-(1,1-dimethylethyl)-2-methyl-1,3-propanediyl ester, since, although no report of any olfactory descriptor associated with this compound was found, it is reported as one of the main volatile compounds in cheese produced with Holstein milk [51]. Descriptors like herbaceous or bell pepper are reported as characteristic attributes of Cabernet Sauvignon wines [52] and methoxypyrazines are mainly responsible for these notes [53]. However, in the present study, it was not possible to identify any methoxypyrazine, probably due to their very low concentration because of the ripeness degree of the grapes since this factor, in addition to others such as ripening temperature and exposure to sunlight, influence the presence and concentration of these compounds [54]. Therefore, both the herbaceous and bell pepper notes could be associated to nonanal, a compound that can provide "green" notes [55]. In the case of yeasty notes, they could be attributed to the fact that neither of the two experimental wines underwent filtration, only cold clarification.

In the case of the wine RWCH, it was associated with odor attributes such as prune, butter, berries, woody, spicy, smoky, and vanilla, as well as flavor attributes such as smoky, vanilla, spicy, woody, and prune. Several of these sensory attributes agree with some volatile compounds associated with this wine. For example, the odor and flavor of prune notes could be due to the β-damascenone wine content since this compound has been reported as having, among other notes, a strong odor of prunes [56]. In addition, (*cis*)-oak-lactone is likely responsible for both the odor and flavor descriptors "woody" and "smoky", two of the notes that are related to this compound extracted from oak wood [57].

On the other hand, the two commercial wines, C1 and C2, had a more fruity and floral profile than the experimental and the control rosé wines, which could be considered a traditional sensory profile for young rosé wine. These wines were linked to odor descriptors like peach, pear, citric, quince, and floral. Regarding flavor descriptors, the wine C1 was more

related to attributes of sweetness and balance, also to notes such as floral, apple, and fruity. Meanwhile, wine C2 was more related to notes of citric, alcohol, strawberry, raspberry, pear, and sulfurous. Floral notes could be due to 2-phenylethyl acetate or 1-dodecanol in the case of wine C2, which were reported as having a rose-like odor [58,59], and to ethyl hexanoate in the case of wine C1, a compound that also can confer flowery notes [60]. Pear and apple notes could be associated with ethyl octanoate, which has an apple-like [61] and pear aroma [60] since this compound was associated with wine C1, as shown in Figure 2. In the case of wine C2, the descriptors of pear could be due to hexyl acetate, a compound that can also confer this note [62]. The odor and flavor citric attributes of wine C2 could be associated with β-linalool, a compound reported as conferring citrus notes [63]. Diethyl succinate and naphthalene, 1,2-dihydro-1,1,6-trimethyl- could be responsible for the flavor and odor floral descriptors since they were reported as one of the several molecules responsible for varietal floral odors in some young wines [57]. Ethyl decanoate, ethyl-9-decenoate, and diethyl succinate could be associated with fruity descriptors [57] in the two commercial wines, although both strawberry and raspberry descriptors in wine C2 could also be attributed to their benzaldehyde content since, depending on the concentration, this compound could confer fruity or berries aromas [64].

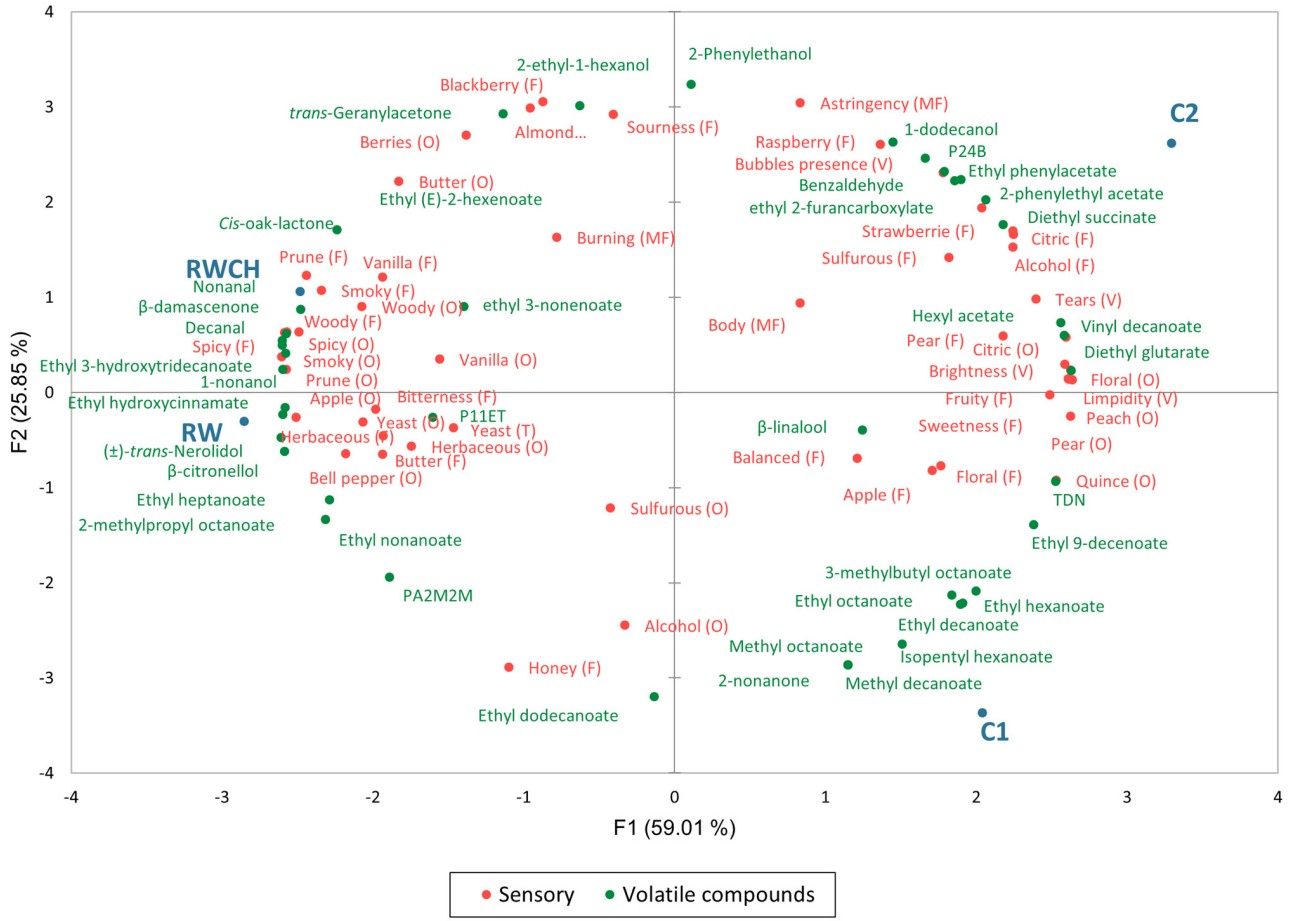

**Figure 2.** Principal component analysis of volatile compounds and sensory attributes of experimental and commercial rosé wines. V = Visual attribute; O = Olfactive attribute; F = Flavor attribute; MF = Mouthfeel attribute; RWCH = Rosé wine matured with oak chips; RW = Rosé wine matured without oak chips; C1 = Commercial rosé wine 1; C2= Commercial rosé wine 2; P11ET = 1-(1-ethoxyethoxy)-pentane; PA2M2M = Propanoic acid, 2-methyl-, 1-(1,1-dimethylethyl)-2-methyl-1,3-propanediyl ester; P24B = Phenol, 2,4-bis(1,1-dimethylethyl)-; TDN = Naphthalene, 1,2-dihydro-1,1,6-trimethyl-.

The great differences existing between the wines that did not have contact with the chips (RW, C1, and C2), and mainly the RW with the two commercial wines, can be attributed to different factors that influence both the sensory and volatile profile of wines, such as the yeast strain and the temperature used in fermentation, the degree of grapes ripeness, the soil and climate in which they were grown, or the winemaking technique [65].

Figure 3 shows the attributes that mainly differentiated the sensory profile of RWCH wine from RW wine. As can be seen, the RWCH wine had a greater intensity in the olfactory descriptor "woody", in the flavor descriptors "vanilla", "smoky", "spicy", and "balanced", and in the mouthfeel attribute "body". It showed a greater presence of the characteristic attributes of wines that have undergone barrel maturation. On the other hand, the RW had a greater intensity in the olfactory descriptors "herbaceous" and "bell pepper" as well as in the flavor descriptors "floral" and "apple", all of them being varietal notes characteristic of young wines. In the case of the commercial wines, they were very similar to each other in terms of these attributes and both were rated with a higher intensity in the flavor descriptors "apple" and "floral", in comparison with the experimental wines, mainly with the RWCH wine.

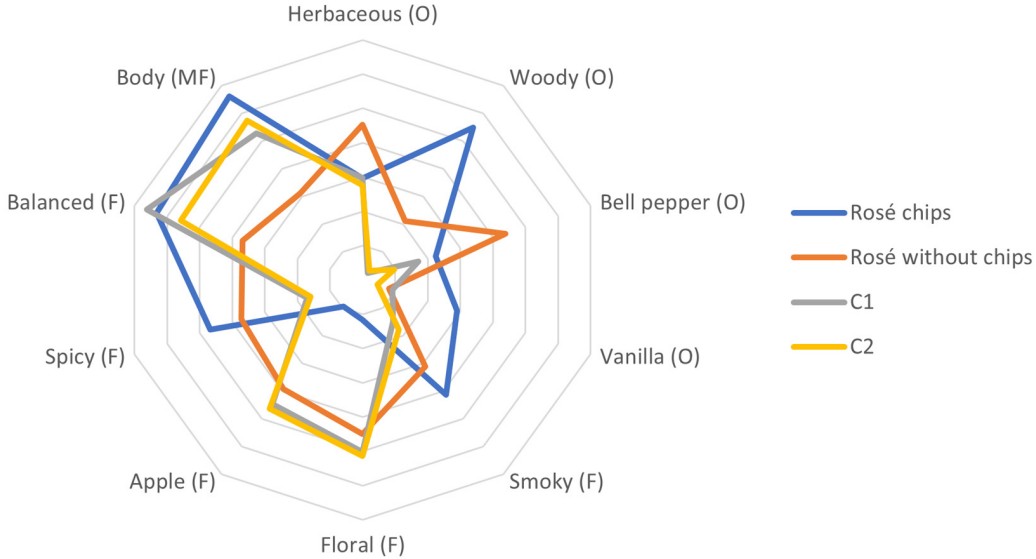

**Figure 3.** Radial graph of the attributes that differentiated the sensory profiles of the experimental wines. O = Olfactive attribute; F = Flavor attribute; MF = Mouthfeel attribute.

As previously described, the addition of wood chips provided the rosé wine with several sensory descriptors associated with wood aging. However, it would be interesting to carry out a more complete investigation in which different factors, such as number of chips, level of toasting, contact time of chips with wine, and even chips shape, were involved, since different effects on the physicochemical, sensory, and volatile profile of wines have been reported [9–11]. For example, Koussissi et al. [9] reported a higher concentration of furfural and *cis*-oak-lactone in a red wine from the Greek grape cultivar Aghiorghitiko that was aged with medium-roasted chips compared to another that had been matured with deep-roasted chips. In terms of the sensory aspect, the former was mostly related to descriptors such as "woody", "smoky", and "spicy", while the latter was more associated with the descriptors "nutty", "earthy", and "bitter". In another investigation on a Chardonnay wine with oak-chips aging (non-toasted and light toasted) for 1, 2, and 3 months, Stegăruș et al. [11] reported a higher concentration of both total lactones and total volatile phenols in the wine aged for 1 month with non-toasted chips compared to wines aged for 2 and 3 months. And, although the wines aged with light toasted chips had a lower concentration of the aforementioned compounds than those aged with unroasted chips, the extraction tendency of these was similar.

## 4. Conclusions

The experimental rosé wines complied with the parameters established by Mexican regulations and their sensory and volatile profiles differed from those of commercial wines, whose profiles were quite similar to conventional rosé wines. Meanwhile, the profiles of the experimental rosé wines comprised some attributes typical of the Cabernet Sauvignon variety and young rosé wines, such as berries, prune, bell pepper, and herbaceous notes. In addition, the red wine matured with oak chips also had characteristic descriptors of wines matured in oak barrels, such as vanilla, woody, smoky, and spicy. Therefore, the maturation of rosé wines in the presence of oak chips could be a good alternative to provide unusual notes in this type of wine and thus offer new alternatives to wine consumers. This study represented the first approach to studying a product that could be an alternative to commercial rosé wines. However, it would be interesting to carry out a more in-depth analysis that includes different maturation times with oak chips and, in addition to an analysis with a trained panel, a consumer analysis to evaluate the acceptance of the product. Also, it would be worth going deeper into the identification and quantification of a greater number of volatile compounds, including varietals, fermentation, and maturation compounds.

**Author Contributions:** M.Á.H.-C.: conceptualization, methodology, formal analysis, investigation, writing—original draft preparation, writing—review and editing. J.R.V.-C.: conceptualization, methodology, resources, writing—review and editing, supervision. H.B.E.-B.: conceptualization, methodology, resources, writing—review and editing, supervision. A.P.-Á.: conceptualization, methodology, resources, supervision. All authors have read and agreed to the published version of the manuscript.

**Funding:** This research received no external funding.

**Data Availability Statement:** All data are available from the authors on request.

**Conflicts of Interest:** The authors declare no conflict of interest.

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
