# Peer review of "Effect of Maturation with American Oak Chips on the Volatile and Sensory Profile of a Cabernet Sauvignon Rosé Wine and Its Comparison with Commercial Wines"

_beverages, doi:10.3390/beverages9030072_

Round 1
Reviewer 1 Report
The article is interesting, but the research methodology and content is too basic, along with the following problems:
1. insufficient discussion in the background about the adequate necessity of the research content;
2. failure to conduct in-depth research or interpretation about the quantitative relationship between oak chips and wine;
3. Line 314 and 330, remove the outer frame lines from Figures 2 and 3.
Author Response
- insufficient discussion in the background about the adequate necessity of the research content;
L. 55-58 A paragraph containing a deeper justification of the necessity for exploring oak chips maturation of rosé wines was added to the text.
2. failure to conduct in-depth research or interpretation about the quantitative relationship between oak chips and wine;
L.401-417. A deeper discussion about the relation between oak chips and wine was added.
3. Line 314 and 330, remove the outer frame lines from Figures 2 and 3.
The outer frame lines from Figures 2 and 3 were removed
Reviewer 2 Report
Dear Authors,
The presented topic is interesting and useful for the wine industry.
There are a few points that should be addressed by the authors:
- First of all, the title of the article is too general. It could be reformulated to highlight the results of your study
Line 69-73: ”Therefore, considering the search for new flavors by an important sector in wine consumption and the advantages offered using oak chips in wine maturation, the aim of this research was to contribute to the limited information of aged rosé wines by studying both volatile and sensory profiles of a varietal rosé wine matured with oak wood chips.” – in the aim of the article it is not specified that the experimental wines will be compared with other wines – it must reformulate the aim of the paper
Subchapter 2.1.1. – the author must indicate the geographical coordinates of the vineyard, climatic conditions of the winegrowing area, the climatic conditions of the vintage year, the training system of the vines, the pruning system, and vineyard management – all these factors have a direct influence on harvest and wine quality
Subchapter 2.1.2. – the authors must provide the source of the oak chips, the toasting level of the chips, the dimensions of the chips
Chapter 2.2.- the authors must provide short descriptions of the basic wine analyses, not only indicate a reference
In Chapter 2.2. and Chapter 2.3. it is not indicated that the analyzes were made in duplicate or triplicate
Table 1. – the authors must indicate the statistical test used if there were duplicate/triplicate, the meaning of the letters
Chapter 2.4. C1 and C2 are rose wines from the Cabernet variety. Are rose wines obtained using the same technology as in the study? C1 and C2 are from the same vintage?
In Table 1 there could be exposed the results for C1 and C2
In Table 2 - the authors must indicate the statistical test used if there were duplicate/triplicate, the meaning of the letters
Figure 3 – why there are not present C1 and C2 in this figure? Since there are the results for both wines?
The authors must verify all over the paper the writing for SO2, using the Italic fonts for Latin words, and so on!
Thank you!
Author Response
There are a few points that should be addressed by the authors:
- First of all, the title of the article is too general. It could be reformulated to highlight the results of your study
The title was reformulated as recomended.
Line 69-73: ”Therefore, considering the search for new flavors by an important sector in wine consumption and the advantages offered using oak chips in wine maturation, the aim of this research was to contribute to the limited information of aged rosé wines by studying both volatile and sensory profiles of a varietal rosé wine matured with oak wood chips.” – in the aim of the article it is not specified that the experimental wines will be compared with other wines – it must reformulate the aim of the paper
L. 75-76. The comparison of the experimental wines with commercial wines was added to the aim of the article.
Subchapter 2.1.1. – the author must indicate the geographical coordinates of the vineyard, climatic conditions of the winegrowing area, the climatic conditions of the vintage year, the training system of the vines, the pruning system, and vineyard management – all these factors have a direct influence on harvest and wine quality
L.87. Vineyard geographical coordinates were added. The remained information was requested to the Vinyard winemaker. However, we have not received any response.
Subchapter 2.1.2. – the authors must provide the source of the oak chips, the toasting level of the chips, the dimensions of the chips
L.102-104. The requested information was added.
Chapter 2.2.- the authors must provide short descriptions of the basic wine analyses, not only indicate a reference
L.108-180. A brief description of the wine analyses was included.
In Chapter 2.2. and Chapter 2.3. it is not indicated that the analyzes were made in duplicate or triplicate
L.109 and L.182. The requested information was added.
Table 1. – the authors must indicate the statistical test used if there were duplicate/triplicate, the meaning of the letters
L.237-238. The requested information was added.
Chapter 2.4. C1 and C2 are rose wines from the Cabernet variety. Are rose wines obtained using the same technology as in the study? C1 and C2 are from the same vintage?
L.205-206. The required information from the commercial wines was added.
In Table 1 there could be exposed the results for C1 and C2
It would be interesting to include the required information but, unfortunately, those analyses were not performed on commercial wines.
In Table 2 - the authors must indicate the statistical test used if there were duplicate/triplicate, the meaning of the letters
L.273-274. The requested information was added.
Figure 3 – why there are not present C1 and C2 in this figure? Since there are the results for both wines?
Commercial wines information was not presented in Figure 3 because we want to highlight the main sensory effect of oak chips maturation.
The authors must verify all over the paper the writing for SO2, using the Italic fonts for Latin words, and so on!
All manuscript was revised and corrected.
Reviewer 3 Report
Abstract:
Line 18 and line 78 (Materials and Methods): Change “Then, their volatile and sensory profiles were performed…” to “
Then, their volatile and sensory profiles were analyzed..”
Line 27: To new wine consumers only? Ad what about traditional consumers?
Introduction
Line 45: I don’t totally agree with this definition (“Rosé wine can be defined as the one that comes from red grapes or a mixture of red 45 and white grapes, fermented in the absence of solid parts”)…Part of the fermentation (the first hours or the first days) occurs in presence of the solid parts. The same authors better explain this concept in the next lines.
Materials and Methods
2.1.1. Variety and origin of the grapes
Why the choice of Cabernet sauvignon grapes? Please, explain.
Line 90: Please, explain the use of a Saccharomyces bayanus yeast strain.
Line 96: Please, explain the use of 3g/L of white oak chips. Why this quantity? And why the choice of this contact time?
Lines 144-145: Why the choice of these two commercial wines?
Line 147: Why the choice f a 15 cm continuous line? It is unusual.
Results and Discussion
From the results, RW (Rosé wine without chips) was strongly different from the commercial wines selected.
Line 241: …is the most abundant 241 acyl-CoA in an alcoholic Saccharomyces cerevisiae fermentation… And what about Saccharomyces bayanus?
Line 278: Authors should explain the absence of methoxypyrazines.
Author Response
Line 18 and line 78 (Materials and Methods): Change “Then, their volatile and sensory profiles were performed…” to “
Then, their volatile and sensory profiles were analyzed..”
L.19 and L.82. The statements were corrected as sugested.
Line 27: To new wine consumers only? Ad what about traditional consumers?
L.27-28. The idea was also extended to traditional wine consumers.
Introduction
Line 45: I don’t totally agree with this definition (“Rosé wine can be defined as the one that comes from red grapes or a mixture of red 45 and white grapes, fermented in the absence of solid parts”)…Part of the fermentation (the first hours or the first days) occurs in presence of the solid parts. The same authors better explain this concept in the next lines.
The definition of rosé wine was conditioned to countries' regulations to avoid generalizing.
Materials and Methods
2.1.1. Variety and origin of the grapes
Why the choice of Cabernet sauvignon grapes? Please, explain.
L.88-89. The reason of the choice of Cabernet Sauvignon variety was included.
Line 90: Please, explain the use of a Saccharomyces bayanus yeast strain.
L.95-97. The use of the yeast was described.
Line 96: Please, explain the use of 3g/L of white oak chips. Why this quantity? And why the choice of this contact time?
L.106-107. The quantity and contact time of the chips were elected according to previous studies on wine.
Lines 144-145: Why the choice of these two commercial wines?
L.205-206. The commercial wines were elected based on their easy availability in Mexican liquor stores.
Line 147: Why the choice f a 15 cm continuous line? It is unusual.
L.209-210. Because the usage of an unstructured 15-cm-scale is commonly applied in Quantitative Descriptive Analysis (QDA). A reference was included.
Results and Discussion
From the results, RW (Rosé wine without chips) was strongly different from the commercial wines selected.
A paragraph including probable factors responsible for that strong difference was added.
Line 241: …is the most abundant 241 acyl-CoA in an alcoholic Saccharomyces cerevisiae fermentation… And what about Saccharomyces bayanus?
L.304-306. Information about S. bayanus was added.
Line 278: Authors should explain the absence of methoxypyrazines.
L.341-344. An explanation about methoxypyrazines absence was included in the manuscript.
Round 2
Reviewer 1 Report
The manuscript has been refined in accordance with personal recommendations, and I personally believe it is ready for journal acceptance.
Author Response
Thanks for your review. Your comments helped to improve the manuscript.
Reviewer 2 Report
Dear Authors,
Thank you for reconsidering your paper. The new title of the study reflects better your results.
Also, some of the points still are to be fixed:
Lines 71-76: what is the purpose of comparing the RWCH and RW with the two commercial wines C1 and C2?
Subchapter 2.1.1. – the authors should also precise the climatic conditions of the winegrowing area, the climatic conditions of the vintage year, the training system of the vines, the pruning system, and vineyard management – all these factors have a direct influence on harvest and wine quality
In Table 1 there could be exposed the results for C1 and C2.
Figure 3 – why there are not present C1 and C2 in this figure? Since there are the results for both wines, regarding the volatile profile. Also, in subchapter 2.4. Sensorial analysis – is specified that the sensorial analysis is made for all 4 wines (RWCH, RW, C1, and C2).
Thank you!

Author Response
Lines 71-76: what is the purpose of comparing the RWCH and RW with the two commercial wines C1 and C2?
L.75-77. The purpose of comparing the experimental wines with commercial wines was included in the manuscript.
Subchapter 2.1.1. – the authors should also precise the climatic conditions of the winegrowing area, the climatic conditions of the vintage year, the training system of the vines, the pruning system, and vineyard management – all these factors have a direct influence on harvest and wine quality
L.89-93. The required information was included.
In Table 1 there could be exposed the results for C1 and C2.
L.243. The required data were included in Table 1.
Figure 3 – why there are not present C1 and C2 in this figure? Since there are the results for both wines, regarding the volatile profile. Also, in subchapter 2.4. Sensorial analysis – is specified that the sensorial analysis is made for all 4 wines (RWCH, RW, C1, and C2).
L.435. Information of C1 and C2 was included in Figure 3, as suggested.
Thank you for your comments. Your recomendatios helped to improve the manuscript.
Reviewer 3 Report
The manuscript has been improved.
I ask the authors to specify the reason of the choice of S. bayanus.
Then, I saw that Methods have been described too in depth. The bibliographic reference is enough, therefore the description should be limited to the difference from the cited methods.
Author Response
I ask the authors to specify the reason of the choice of S. bayanus.
L.101-102. The reason of using S.bayanus was specified.
Then, I saw that Methods have been described too in depth. The bibliographic reference is enough, therefore the description should be limited to the difference from the cited methods.
The description of the methods was included at the request of one of the reviewers.
Thank you for your comments. Your suggestions helped to improve the manuscript.